# Recipient-Reported Reactogenicity of Different SARS-CoV-2 Vaccination Regimens among Healthcare Professionals and Police Staff in Germany

**DOI:** 10.3390/vaccines11071147

**Published:** 2023-06-25

**Authors:** Katharina Rau, Edgar von Heeringen, Nina Bühler, Stefan Wagenpfeil, Sören L. Becker, Sophie Schneitler

**Affiliations:** 1Center for Infectious Diseases, Institute of Medical Microbiology and Hygiene, Saarland University, 66421 Homburg, Germany; kathi.rau97@gmail.com (K.R.); soeren.becker@uks.eu (S.L.B.); 2Landespolizeipräsidium—LPP 33, 66121 Saarbrücken, Germany; 3Institute for Medical Biometry, Epidemiology and Medical Informatics, Saarland University, 66421 Homburg, Germany; 4Bethanien Hospital, Clinic of Pneumology and Allergology, Center for Sleep Medicine and Respiratory Care, Institute of Pneumology at the University of Cologne, 42699 Solingen, Germany

**Keywords:** COVID-19, homologous vaccination, heterologous vaccination, ChAdOx1 nCoV-19, BNT162b2, mRNA-1273, vaccine hesitancy, healthcare staff

## Abstract

The rapid availability of effective vaccines against SARS-CoV-2 was key during the COVID-19 pandemic. However, vaccine hesitancy and relatively low vaccine coverage rates among the general population and particularly vulnerable populations such as healthcare staff reduced the potential benefits of these vaccines. During the early phase of the pandemic, fear of vaccine-related adverse events was common among individuals who refused vaccination. Between March and May 2021, we comparatively assessed the self-reported reactogenicity of different SARS-CoV-2 prime-boost regimens using mRNA-based (BNT162b2 and mRNA-1273) and vector-based vaccines (ChAdOx1 nCoV-19) in (a) healthcare workers (HCW), and (b) police staff from southwest Germany. The majority of participants (71.8%; 1564/2176) received a homologous vaccination. Among HCW, 75.0% were female, whereas 70.0% of police staff were male. The most frequently reported reactions following the first vaccine administration were pain at the injection site (77.94%; 1696/2176), tiredness (51.75%; 1126/2176), and headache (40.44%; 880/2176), which were more commonly reported by HCW as compared to police staff. In homologous, mRNA-based and heterologous vaccination schedules, more reactions were reported after the second vaccine dose. We conclude that the frequency and intensity of self-perceived vaccine reactogenicity may differ between specific population groups and might be mitigated by tailored communication strategies.

## 1. Introduction

Since the emergence of SARS-CoV-2 and the onset of the COVID-19 pandemic in early 2020 [1,2], it became rapidly evident that the development of effective vaccines would represent one of the major goals to reduce the spread and the burden on healthcare systems due to COVID-19. Facilitated by decades of basic research pertaining to different vaccination modalities, several vaccines were authorized for human use within approximately one year of the first COVID-19 cases [3].

One of the main reasons for the rapid development and introduction of a vaccine was the lack of an effective drug against COVID-19 and the lack of any pre-existing immunity to this novel pathogen at the global level [4]. Dexamethasone was one of the few drugs that was shown to be effective early on and had a positive effect on reducing mortality. The greatest effect was achieved in mechanically ventilated patients with a disease duration of more than seven days [5].

Other reasons for the rapid development of a vaccination strategy included economic and social constraints. As a result of the COVID-19 hygiene precautions and infection prevention strategies, “lockdowns” were established in many countries, which gave rise to many unintended consequences, such as reduced healthcare system contacts, with detrimental effects on cancer screening and early treatment of malignant and infectious diseases, for example [6].

In Europe, two major principles were mainly used, i.e., mRNA-based vaccines such as BNT162b2 from the companies Pfizer/BioNTech (B) [7] or mRNA-1273 from Moderna (M) [8], and vector-based vaccines such as ChAdOx1 nCoV-19 from AstraZeneca/Oxford (A) [8]. In the first studies, these vaccines provided excellent protection against symptomatic and severe SARS-CoV-2 infection, COVID-19-related hospitalization, and death [7,9]. Later in the pandemic, with the emergence of new virus variants such as Omicron [10], protection against asymptomatic and mild infections was weaker [11,12], but vaccination still provided effective protection against severe COVID-19 [12,13].

The original dosing schedules recommended the application of two vaccine doses at least 3–4 weeks apart, and these were later modified to include additional doses to provide longer-term protection [14]. Use of the vector vaccine ChAdOx1 nCoV-19 was soon linked to the rare occurrence of thrombotic events, mainly leading to cerebral venous sinus thrombosis [15], which also led to the introduction of heterologous vaccine recommendations, e.g., boosting with mRNA-based vaccines after vector-based priming. Such regimens were shown to be highly immunogenic but also showed differences with regard to the reported reactogenicity [16,17].

The general attitude of an individual towards vaccination depends on different aspects, such as trust in the vaccine itself and its safety, fear of side effects, or the income of the country [18,19]. Furthermore, willingness to vaccinate can be influenced on the one hand by fear of acquiring the respective disease (e.g., COVID-19) or fear of infecting relatives [19]. Worldwide, women were, in general, reported as more likely to have a negative attitude towards COVID-19 vaccination due to concerns of thrombotic events or a suspected potential for adverse sequelae on future pregnancies [18]. In contrast, among healthcare workers (HCW), men have a more negative attitude toward vaccinations than women [19]. Moreover, physicians are more likely to get vaccinated than nurses [20]. Many countries have developed strategies on how vaccination should be rolled out in system-relevant areas, sometimes also called “employees of the critical infrastructure”. As is well known, the health sector has been considered in all countries, but also other sectors such as the fire brigade or police staff. Regarding the vaccination coverage of other system-relevant occupational groups, such as employees of police stations, there are no evidence-based findings or studies available thus far. However, little research has been carried out to investigate whether the acceptance and self-reported vaccine reactions differ between different professional groups. To this end, we conducted a questionnaire-based survey to assess and compare the reported reactogenicity of homologous and heterologous COVID-19 vaccination among HCW and police staff in southwest Germany during an early phase of the vaccination campaign (March–May 2021).

## 2. Materials and Methods

### 2.1. Study Sites

The overall goal of this study was to assess and describe the reactogenicity of SARS-CoV-2 vaccines among HCW as compared to police staff, using self-reported questionnaire data. The study among healthcare staff was integrated into the SARS-CoV-2 vaccination campaign carried out at Saarland University Medical Center in Homburg/Saar, Germany, between 3 March and 4 May 2021.

Police staff were recruited in the vicinity to this hospital (max. distance: 38 km) during a specific vaccination campaign between 10 May and 14 May 2021.

### 2.2. Design and Data Collection Procedures

Information about the study and an invitation to participate were mainly distributed through an e-mail-based newsletter at Saarland University Medical Center, which reached medical staff, administrative employees, as well as technical and all other employees. Furthermore, a web-based tool was used to book vaccination appointments (samedi GmbH, Berlin, Germany); during the booking process, all individuals were also asked whether they would be interested in participating in this study. The study information sheet was sent along with the booking confirmation via e-mail. The actual study inclusion was done in a face-to-face interview on the day of the first vaccination. Further communication with participants, e.g., about reminders to fill in the questionnaire at different timepoints, was via e-mail.

For the recruitment of police staff, employees were enrolled in the study at a nearby police department by the responsible police physician. All individuals were informed about the study procedures on site. Once informed consent was obtained, the signed sheets were forwarded to the study team, and this cohort also received the questionnaires and further reminders by e-mail.

### 2.3. Questionnaire

The questionnaire was designed based on known common and less common side effects of SARS-CoV-2 vaccines described in the pivotal studies. The questionnaire was independently shown to several regional experts in the field of vaccination and immunology, and a pre-test was carried out before wider use. It was divided into two parts, i.e., (1) local reactions such as swelling or redness at the injection site, and (2) systemic side effects such as fever, lymphadenopathy, or gastrointestinal complaints, to allow for a structured assessment of the reactions. Each participant was able to tick “yes” or “no” for each side effect. In case of “yes”, the participant was asked to rate the intensity of the reaction on a grading scale, as follows: (i) mild reaction (rated as 1 or 2); (ii) moderate reaction (rated as 3 or 4); and (iii) severe reaction (rated as 5 or 6).

Additionally, the questionnaires included basic epidemiological data such as age and sex, past medical history, current medication as well as the type of vaccine used.

Each participant filled in the questionnaire (1) five days after the first vaccination, (2) five days after the second injection, and (3) 40 days after the second vaccination. In case of any severe vaccination side effects beyond the known “normal” range, feedback was given to the occupational physicians of the Saarland University Medical Center or the respective police physicians, who would then follow up on these incidents with the responsible regulatory authorities.

All questionnaires were filled in using a paper-based format and could be delivered digitally (e-mail), by fax, or by regular mail.

### 2.4. Statistical Analysis

The data were managed and analyzed using Excel (version 2016) as well as IMV-SPSS Statistics Version 28.0.1.0 (142; IBM Deutschland GmbH; Ehningen, Germany).

The goal of this study was to analyze and comparatively assess the vaccination-associated reactogenicity among HCW and police staff. To this end, multiple comparisons between the two groups were carried out using ordinal or binary logistic regression to account for possible confounding with respect to age and gender, respectively. Outcome variables were specified as dependent, while group, age, and gender were considered to be independent variables for regression approaches. Any *p*-values are two-sided and subject to a significance level of 5%. We did not account for the issue of multiple statistical testing due to the explorative nature of the study. Hence, raw, unadjusted *p*-values are reported.

## 3. Results

### 3.1. Study Profile and Epidemiological Characteristics

Following the initial recruitment of 3176 individuals, the final study cohort comprised 2176 participants after excluding individuals who either did not fill in the questionnaire surveys or did not fulfill all inclusion criteria (Figure 1). The majority of participants received a homologous vaccination, mainly with B (N = 1387), followed by vaccines A (N = 171) and M (N = 6). Heterologous vaccination was less frequently employed and comprised initial priming with the A vaccine (n = 612), which was either followed by B (N = 394) or M (N = 218).

Among HCW, there were more female than male participants (1212 vs. 405), while an opposite pattern was observed for police staff, i.e., 391 male vs. 168 female participants. The mean age was 39.5 years in the HCW group and 48.1 years among police members. Details on the sex and age distribution are given in Figure 2.

### 3.2. Local Vaccine Reactions in HCW and Police Staff

Among all vaccinees, local pain at the injection site was the most frequently reported reaction after the first vaccination, with a prevalence of 82.4% in homologous B/B vaccination, 70.8% in homologous A/A vaccination, and 80.3% for heterologous vaccination (A/B or A/M). Except for local pain at the injection site, B/B vaccinees reported more local reactions following the second vaccine dose. In contrast, A/A vaccinated individuals experienced fewer vaccine reactions after the second dose. In heterologous vaccination, boosting with either B or M was associated with higher levels of local reactions than homologous A/A vaccination. Further common reactions were local swelling, redness, and itching.

In general, police employees reported pain after vaccinations less frequently than HCW. However, they described redness at the injection site after the first dose more commonly than HCW (10.3% vs. 5.4%). Although less pain was reported in terms of absolute numbers, police staff described more intense pain (according to a semi-quantitative grading scale) compared to HCWs, with 9.5% reporting severe pain compared to 4.5% in the HCW group. Details are displayed in Figure 3.

Considering potential confounding due to differences in age and sex, binary logistic regression revealed a statistically significant difference only in risk of the overall reported vaccine reactions after the first homologous B/B vaccination between HCW and police staff (*p*-value < 0.001), but neither for the overall vaccination procedure (i.e., considering first and second vaccine application together; *p*-value 0.70) nor for the second vaccination alone (*p*-value 0.91).

Confounder analyses yielded significant positive gender and negative age effects, i.e., female gender and higher age were associated with a significantly higher and lower risk of vaccine reactions, respectively. This held consistently true for reported vaccine reactions after the first and second homologous B/B vaccination and for the overall vaccination procedure (i.e., considering the first and second vaccine application together).

### 3.3. Systemic Vaccine Reactions in Healthcare Workers and Police Staff

In general, HCW reported more reactions after both vaccinations. Most reactions among HCW were recorded in the heterologous vaccination groups, followed by B/B and A/A. Reactions following the first vaccination were more common in the A/A group, while individuals receiving B/B reported more frequent complaints after the second vaccination.

In both groups (HCW and police staff), individuals having received the B/B vaccination reported short-lasting fatigue as the most common reaction after both vaccinations. However, in comparison, HCW reported fatigue after the second vaccination more frequently than police officers (59.5% vs. 48.4%) and graded this fatigue as more severe (11.6% vs. 8.2%). Another common reaction after receiving the second shot among homogenous B/B immunization was headache (27.8% vs. 21.9%). HCW were more likely to complain of nausea after the second vaccination than the police workers (30.3% vs. 5.6%). Other common systemic side effects, especially after the second vaccination, included aching limbs (25.4% HCW vs. 19.8% police staff) or malaise (32.2% HCW vs. 19.7% police staff).

In the homologous A/A group, chills and fever occurred more frequently after the administration of the first dose as compared to the second dose. In heterologous A/M vaccination schedules, fatigue occurred most frequently during the initial vaccination (75.2%). After the second vaccination with M, 75.7% likewise reported fatigue, as compared to 59.4% of individuals vaccinated with B. Details are shown in Figure 4.

Using ordinal logistic regression, a statistically significant difference was observed for the occurrence of fever after the first vaccination between both groups (*p*-value 0.03). There were no statistically significant differences between HCW and the police with respect to lymph node (LN) swelling, tiredness, insomnia, malaise, headache, muscle pain, joint pain, limbs pain, chills, and nausea/vomiting (*p*-values = 0.7, 0.25, 0.36, 0.31, 0.18, 0.65, 0.86, 0.37, 0.13 and 0.48, respectively) after the first vaccine application.

Following the second vaccine application, there were no statistically significant differences between HCW and police staff pertaining to LN Swelling, tiredness, insomnia, malaise, headache (although rather low *p*-values, pointing towards an increased risk for police staff), muscle pain, joint pain, limbs pain, chills, fever (although with a rather low *p*-value, indicating a decreased risk for police staff) and nausea/vomiting (*p*-values = 0.48, 0.55, 0.79, 0.73, 0.09, 0.9, 0.99, 0.61, 0.57, 0.1 and 0.22, respectively). The small number of participants not providing severity of vaccine reactions (grey scale) were excluded from this kind of regression analysis.

In order to account for possible bias due to the exclusion of participants not providing the severity of vaccine reactions (grey scale), we carried out a sensitivity analysis without considering the severity of vaccine reactions. Hence, we used logistic regression approaches with dependent variables as previously; however, we combined the colored and grey scale to one category of the vaccine reaction. The second category was having “no reaction to vaccine application”. In these analyses, the only statistically significant group difference with decreased risk for police staff was fever following the second vaccine application (*p*-value = 0.02).

Confounder analyses using ordinal regression yielded significant positive gender and negative age effects, i.e., female gender and higher age were associated with a significantly higher and lower risk of specific vaccine reactions after the first vaccination with respect to tiredness, malaise, headache, and muscle pain, respectively. For chills and nausea/vomiting, there were only significant positive gender effects, with a higher risk for females. Otherwise, there was no significant age or gender effect. Considering the setting of sensitivity analysis with logistic regression, i.e., ignoring severity grading of specific vaccine reaction, we obtained almost the same results for confounder analyses as in original ordinal regressions for any of the outcome variables, except for muscle pain. Here, the age effect remained significantly negative, whereas the gender effect was no more significant.

Following the second vaccine application, female gender and higher age were associated with a significantly higher and lower risk of vaccine reactions with respect to LN swelling, tiredness, malaise, headache, muscle pain, limb pain, chills, fever, and nausea/vomiting, respectively. For the remaining variable joint pain after the second vaccine application, there was only a significant positive gender effect with a higher risk for females. Considering the setting of sensitivity analysis with logistic regression, i.e., ignoring severity grading of specific vaccine reaction after the second vaccine application, we again obtained almost the same results for confounder analyses as in original ordinal regressions for any of the outcome variables, except for muscle pain. Again, the age effect remained significantly negative, whereas the gender effect was no more significant.

## 4. Discussion

We found that HCW were more likely to report perceived vaccine reactions than police staff. This held true for all analyzed vaccine application schedules, i.e., homologous vaccination with either A, B, or M, and heterologous mRNA-based boosting after vector-based priming with A. Furthermore, the intensity of self-reported reactions was, in most instances, more pronounced in HCW. We also observed vaccine-specific particularities, e.g., homologous vaccination with A led to a higher number of complaints after the first vaccine dose, while mRNA-based vaccination led to more self-reported reactions after the second dose. This finding is in line with many studies from the current literature [21,22]. Of note, the occurrence of self-reported vaccine reactions may also serve as a proxy for the immunogenicity of a given vaccine. Heterologous vaccination with A/B, for example, which had higher rates of self-reported symptoms than homologous A/A vaccination, was also reported to be more immunogenic than A/A vaccination [22,23].

It has been observed worldwide that the willingness to get vaccinated against COVID-19 decreased as the pandemic evolved further. The German “COVIMO” study observed a sharp loss of interest in the ChAdOx1 nCoV-19 vaccine among the general population following the first reports of thrombotic adverse events [24]. However, very few studies have investigated differences between occupational groups with regard to vaccine uptake and the occurrence of self-reported vaccine reactions. One of the few studies addressing COVID-19 was conducted in Poland and elucidated that 58.9% of the professional medical staff perceived the new vaccines as safe, whereas this was only shared by 27% of the general population [20].

More studies pertaining to differences among specific occupational groups have analyzed reasons for and against influenza vaccination among HCW [20,25,26]. A longitudinal survey from Basel, Switzerland, reported that female nurses were less likely to get vaccinated, with fears of unintended long-term consequences and a perceived violation of the right of self-determination being two highly prevalent reasons in this group, which were far less frequently expressed by other HCW groups [27]. Interestingly, group-specific differences seem to prevail across multiple cultural and geographical settings; for example, a recent study among 1476 HCW from Somalia elucidated that COVID-19 vaccine hesitancy was greater among primary healthcare professionals and those with a master’s degree [28]. A French study assessed the acceptability of mandatory HCW vaccination against influenza, measles, pertussis, and varicella in addition to the already compulsory COVID-19 vaccination. The overall self-reported acceptability was highest for measles (73.1%) and lowest for influenza (42.7%) and revealed a major role of age, sex, and HCW category [29].

There is a paucity of data regarding the acceptance of vaccination among other essential service providers beyond HCW. In 2009 and 2010, a British survey assessed the uptake of the pandemic influenza A H1N1 vaccine among police employees. The study found a general willingness to get vaccinated in 39.7%, with a fear of side effects being the most common reason against vaccination [30]. A US American study found considerable differences in COVID-19 vaccine uptake across different essential service providers, with firefighters being less likely to be vaccinated than law enforcement officers [31]. However, we are not aware of previous comparative studies between HCW and police staff, which were both considered as part of the “essential infrastructure” to be prioritized for early vaccination during the COVID-19 pandemic. While we are unable to decipher whether the observed differences with regard to vaccine reactogenicity can be primarily explained by the occupational group or rather by other factors, we believe that future studies should consider differences between occupational groups as this might help to guide the development of tailored vaccine communication strategies. Indeed, it might be speculated that HCW are more likely to report “side effects” after vaccination because they are well-trained to observe, document, and report medical conditions, which might lead to a higher “reporting index” or increased awareness.

Several limitations of our study are offered for consideration. First, our study was monocentric and was conducted during the early phase of vaccination in the pandemic, when general interest, but also fears and doubts surrounding the new vaccines were relatively high in Germany. Second, the focus on HCW and police staff does not allow for drawing any conclusions that are representative of the general population. Third, it might have been instructive to further analyze the distinct professions within the HCW group (e.g., physicians, nurses, laboratory technicians). Fourth, due to the character of the study, we had to rely on participants’ accounts and could not, for example, objectively verify self-reported fever. Fifth, this study was limited to vaccine reactions following the first two shots but did not include any subsequent booster vaccinations.

## 5. Conclusions

Homologous and heterologous vaccination regimens against SARS-CoV-2 elicit different self-reported vaccine reactions, and these are further influenced by the personal background and apparently also the occupational group of vaccinated HCW and police staff. We thus conclude that the frequency and intensity of self-perceived vaccine reactogenicity may differ between specific population groups. Considering these findings might be important to develop and mitigate tailored, target group-oriented communication strategies for future vaccination campaigns. The development of these communication and education strategies should first and foremost involve a panel of experts, including representatives from science, the medical services, and possibly also from the communication science sector. This committee should then make a clear recommendation, which should be implemented accordingly by policymakers. In addition, misinformation should be addressed and corrected as quickly as possible to overcome vaccine hesitancy and potential concerns within the population with evidence-based scientific arguments. However, it should also be considered that especially in system-relevant areas that are not close to the health sector, key persons are also identified in the respective areas who help to create communication channels and build mutual trust in order to implement the envisaged collaboration and outcome.

## Figures and Tables

**Figure 1 vaccines-11-01147-f001:**
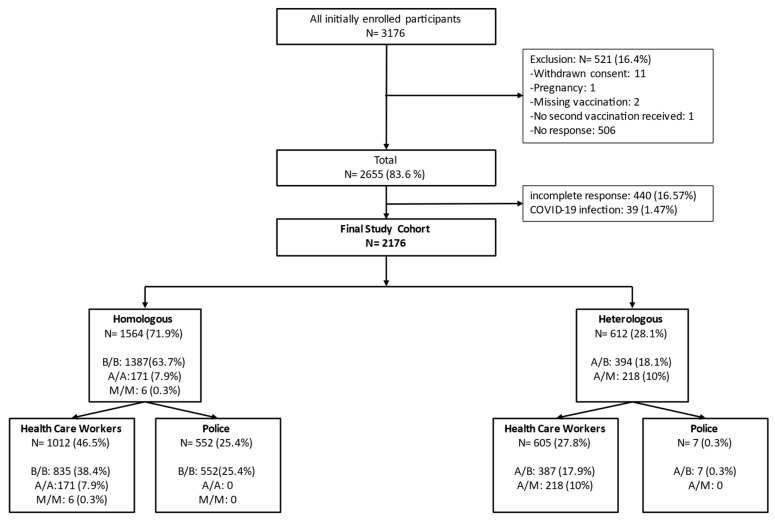
Study flowchart. A = AstraZeneca, B = BioNTech/Pfizer, M = Moderna.

**Figure 2 vaccines-11-01147-f002:**
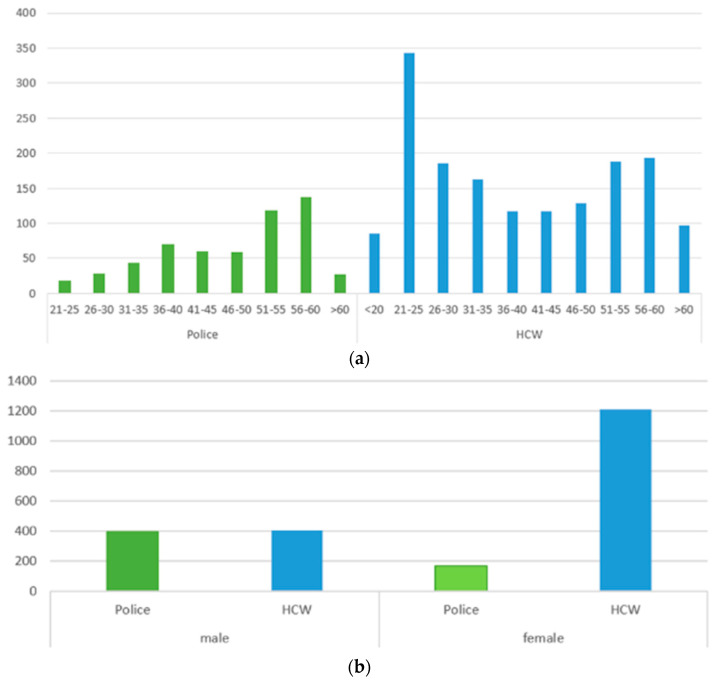
(**a**) Age distribution and (**b**) sex distribution among HCW and police staff in a study assessing the reactogenicity of SARS-CoV-2 vaccination in southwest Germany.

**Figure 3 vaccines-11-01147-f003:**
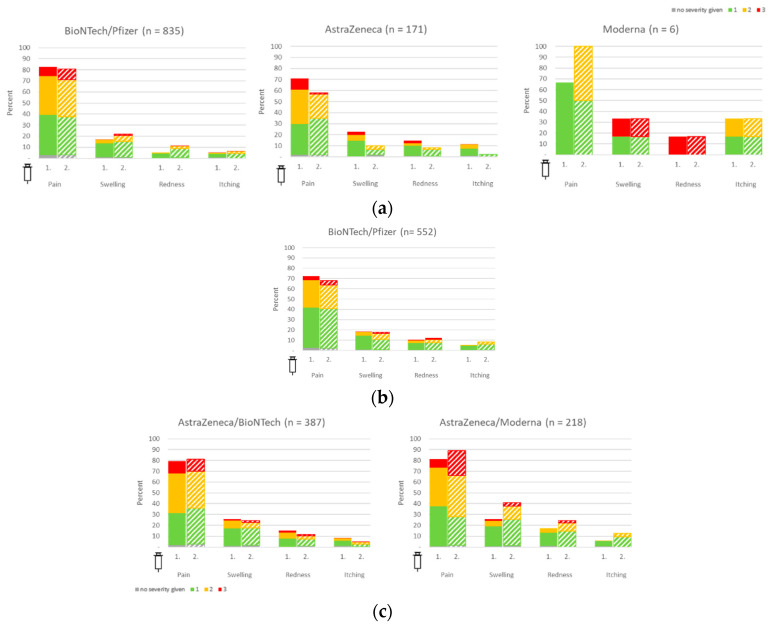
Local vaccine reactions in healthcare workers and police staff. (**a**) Homologous vaccination regimens among HCW; (**b**) homologous vaccination regimens among police staff, (**c**) heterologous vaccination regimens among HCW. Color grading scale: green, light reactions; yellow, moderate reactions; red, severe reactions.

**Figure 4 vaccines-11-01147-f004:**
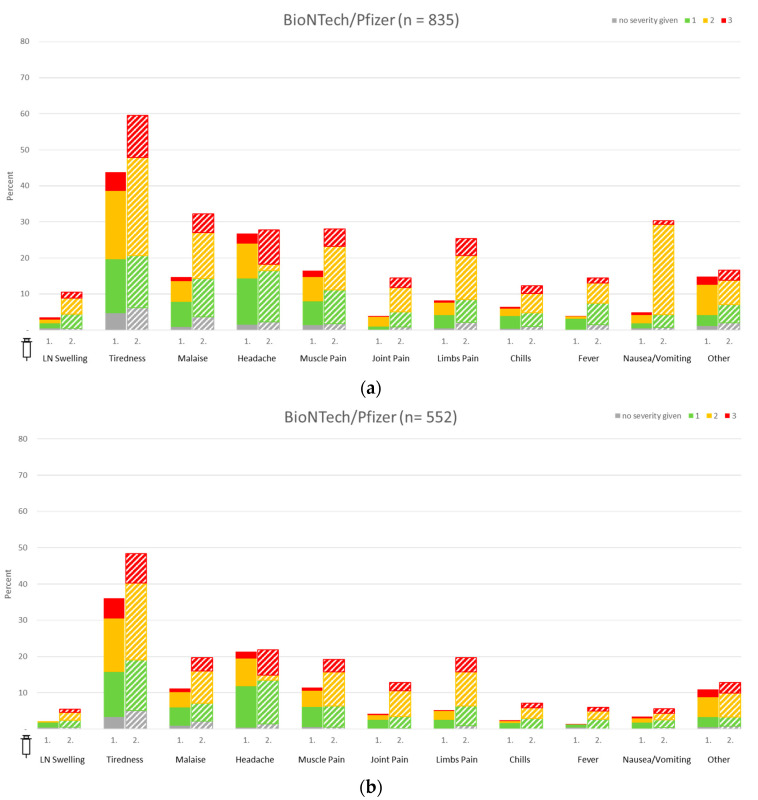
Systemic vaccine reactions in healthcare workers (HCW) and police staff. (**a**) Homologous vaccination regimens among HCW; (**b**) homologous vaccination regimens among police staff. Color grading scale: green, light reactions; yellow, moderate reactions; red, severe reactions; grey, no severity grade given.

## Data Availability

Data can be made available upon request. Please contact the corresponding author.

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
