# Peer review of "Recipient-Reported Reactogenicity of Different SARS-CoV-2 Vaccination Regimens among Healthcare Professionals and Police Staff in Germany"

_vaccines, 2023, doi:10.3390/vaccines11071147_

Round 1

Reviewer 1 Report

In this MS, the authors comparatively assessed the self-reported reactogenicity of different SARS-CoV-2 prime-boost regimens using mRNA-based (BNT162b2 and mRNA-1273) and vector-based vaccines (ChAdOx1 nCoV-19) in healthcare workers (HCW), and police staff from southwest Germany. And they presented a conclusion that the frequency and intensity of self-perceived vaccine reactogenicity may differ between specific population groups and might be mitigated by tailored communication strategies. The results have certain significance in understanding the influencing factors of side effects after vaccination in a large population and developing corresponding response measures.

However, except for differences in profession, there are also significant differences in gender and age distribution between the two populations surveyed and analyzed by the author. Based on the analyzed results, it is difficult to infer which of the background (professional knowledge) or individual objective characteristics has a greater impact on the self-perceived vaccine reactogenicity. The author may need to show the impact of age and gender differences on the self-perceived vaccine reactogenicity across the entire survey population. Further demonstrate whether there are significant differences between objectively observable symptoms such as fever, redness, and swelling, as well as subjective sensory symptoms such as pain and itching, among different populations.

Author Response

In this MS, the authors comparatively assessed the self-reported reactogenicity of different SARS-CoV-2 prime-boost regimens using mRNA-based (BNT162b2 and mRNA-1273) and vector-based vaccines (ChAdOx1 nCoV-19) in healthcare workers (HCW), and police staff from southwest Germany. And they presented a conclusion that the frequency and intensity of self-perceived vaccine reactogenicity may differ between specific population groups and might be mitigated by tailored communication strategies. The results have certain significance in understanding the influencing factors of side effects after vaccination in a large population and developing corresponding response measures.

Response: We thank the Reviewer very much indeed for the detailed analysis and the positive overall evaluation of our manuscript.

However, except for differences in profession, there are also significant differences in gender and age distribution between the two populations surveyed and analyzed by the author. Based on the analyzed results, it is difficult to infer which of the background (professional knowledge) or individual objective characteristics has a greater impact on the self-perceived vaccine reactogenicity. The author may need to show the impact of age and gender differences on the self-perceived vaccine reactogenicity across the entire survey population. Further demonstrate whether there are significant differences between objectively observable symptoms such as fever, redness, and swelling, as well as subjective sensory symptoms such as pain and itching, among different populations.

Response: We thank the reviewer for this important and valuable comment, which we also considered during the analysis. Hence, because of age and gender differences between the group of health care workers (HCW) and police staff, we analysed for confounding in binary logistic regression analyses, i.e. we calculated group effects adjusted for age and gender differences. With respect to results given in Figure 3, we now state in the revised work: “Considering potential confounding due to differences in age and sex, binary logistic regression revealed a statistically significant difference only in risk of the overall reported vaccine reactions after the first homologous B/B vaccination between HCW and police staff (p-value < 0.001), but neither for the overall vaccination procedure (i.e. considering first and second vaccine application together; p-value 0.70) nor for the second vaccination alone (p-value 0.91).”

In addition, we now describe specific age and gender effects for the adjusted regression analyses:

“Confounder analyses yielded significant positive gender and negative age effects, i.e. female gender and higher age were associated with a significant higher and lower risk of vaccine reactions, respectively. This holds consistently true for reported vaccine reactions after the first as well as second homologous B/B vaccination, and for the overall vaccination procedure (i.e. considering first and second vaccine application together).”

Also, we considered possible confounding of age and gender in analyses considering outcomes in Figure 4, i.e. we calculated group effects adjusted for age and gender differences. We now describe specific age and gender effects for the adjusted ordinal and logistic regression analyses, as follows: “Confounder analyses using ordinal regression yielded significant positive gender and negative age effects, i.e. female gender and higher age were associated with a significant higher and lower risk of specific vaccine reactions after the first vaccination with respect to tiredness, malaise, headache and muscle pain, respectively. For chills and nausea/vomiting, there were only significant positive gender effects with a higher risk for females. Otherwise, there was no significant age or gender effect. Considering the setting of sensitivity analysis with logistic regression, i.e. ignoring severity grading of specific vaccine reaction, we obtained the almost the same results for confounder analyses as in original ordinal regressions for any of the outcome variables, except for muscle pain. Here the age effect remains significantly negative, whereas the gender effect is no more significant.

Following the second vaccine application, female gender and higher age were associated with a significant higher and lower risk of vaccine reactions with respect to LN swelling, tiredness, malaise, headache, muscle pain, limbs pain, chills, fever and nausea/vomiting, respectively. For the remaining variable joint pain after second vaccine application, there was only a significant positive gender effect with a higher risk for females.“

Reviewer 2 Report

Thank you for this paper. My suggestions are only minor. These include:

A) Introduction

1) You mention thrombocytopenia and thromboembolism after covid-19 vaccination. As you mention - this led to initial concerns with the Oxford/ AZ vaccine. This may be because of the fast roll-out of this vaccine in the UK with subsequent studies demonstrating similar events with the other vaccines (e.g. Hippisley-Cox J et al. Risk of thrombocytopenia and thromboembolism after covid-19 vaccination and SARS-CoV-2 positive testing: self-controlled case series study. Bmj. 2021;374:n1931 and Abrignani MG  et al. COVID-19, Vaccines, and Thrombotic Events: A Narrative Review. J Clin Med. 2022;11 among others. This further demonstrates the power of limited information without context, etc., in this field as we are very aware of the appreciable impact of misinformation promulgated by social media on vaccine uptake 

2) Key reasons for the need for vaccines was the lack of therapeutic effect of the various proposed re-purposed medicines - apart from dexamethasone - despite all the hype, e.g.  Horby P et al. Effect of Hydroxychloroquine in Hospitalized Patients with Covid-19. N Engl J Med. 2020;383:2030-40; Horby P et al. Dexamethasone in Hospitalized Patients with Covid-19. N Engl J Med. 2021;384:693-704; Lopinavir-ritonavir in patients admitted to hospital with COVID-19 (RECOVERY): a randomised, controlled, open-label, platform trial. Lancet. 2020;396:1345-52; Dyer O. Covid-19: Remdesivir has little or no impact on survival, WHO trial shows. Bmj. 2020;371:m4057. Alongside this, the appreciable economic impact of COVID-19 lockdown measures - as well as the impact of lockdown measures on the identification and management of NCDs including cancers, etc

3) I can understand why HCWs - especially as patients will be guided by them - but why police personnel - good to also explain this further

B) Methodology

1) I see that the questionnaire was based on expert opinion. I believe this is OK so long as this is spelled out and justified (I have used this approach in MDPI Journals including Vaccines and Antibiotics)

2) However - why no pilot study to enhance its robustness

C) Conclusion - you talk about communication strategies - who should instigate these/ follow these up given the appreciable extent of misinformation that exists with the Vaccines - and the need to overcome these? Good to give some guidance here

Author Response

Thank you for this paper. My suggestions are only minor. These include:
Response: We thank the Reviewer for his/her kind words.

  1. A) Introduction

1) You mention thrombocytopenia and thromboembolism after covid-19 vaccination. As you mention - this led to initial concerns with the Oxford/ AZ vaccine. This may be because of the fast roll-out of this vaccine in the UK with subsequent studies demonstrating similar events with the other vaccines (e.g. Hippisley-Cox J et al. Risk of thrombocytopenia and thromboembolism after covid-19 vaccination and SARS-CoV-2 positive testing: self-controlled case series study. Bmj. 2021;374:n1931 and Abrignani MG  et al. COVID-19, Vaccines, and Thrombotic Events: A Narrative Review. J Clin Med. 2022;11 among others. This further demonstrates the power of limited information without context, etc., in this field as we are very aware of the appreciable impact of misinformation promulgated by social media on vaccine uptake.

Response: We thank Reviewer 2 for emphasising this aspect and for the comment. While revising, we highlighted important aspects of miscommunication in the pandemic and during the vaccination campaign even more, and we also included this aspect in the revised ‘Conclusion’.

2) Key reasons for the need for vaccines was the lack of therapeutic effect of the various proposed re-purposed medicines - apart from dexamethasone - despite all the hype, e.g.  Horby P et al. Effect of Hydroxychloroquine in Hospitalized Patients with Covid-19. N Engl J Med. 2020;383:2030-40; Horby P et al. Dexamethasone in Hospitalized Patients with Covid-19. N Engl J Med. 2021;384:693-704; Lopinavir-ritonavir in patients admitted to hospital with COVID-19 (RECOVERY): a randomised, controlled, open-label, platform trial. Lancet. 2020;396:1345-52; Dyer O. Covid-19: Remdesivir has little or no impact on survival, WHO trial shows. Bmj. 2020;371:m4057. Alongside this, the appreciable economic impact of COVID-19 lockdown measures - as well as the impact of lockdown measures on the identification and management of NCDs including cancers, etc.
Response: We fully agree with this comment. Hence, we have included these aspects pertaining to the treatment of COVID-19 and its limited efficacy in the introductory section of our manuscript to put further emphasis on the importance and necessity of the vaccines used.

3) I can understand why HCWs - especially as patients will be guided by them - but why police personnel - good to also explain this further

Response: Thank you for pointing out this slightly unclear aspect. We have explained the motivation to include also police staff further in the revised ‘Introduction’, as follows: “Many countries have developed strategies on how vaccination should be rolled out in system-relevant areas, sometimes also called ‘employees of the critical infrastructure’. As is well known, the health sector has been considered in all countries, but also other sectors such as the fire brigade or police staff. Regarding the vaccination coverage of other system-relevant occupational groups, such as employees of police stations, there are no evidence-based findings or studies available thus far.”

  1. B) Methodology

1) I see that the questionnaire was based on expert opinion. I believe this is OK so long as this is spelled out and justified (I have used this approach in MDPI Journals including Vaccines and Antibiotics)

Response: Thank you very much for your close look at the structure of the questionnaire. Indeed, the questionnaire used in this study is based both on expert opinion and on the reported spectrum of side effects described in the respective pivotal studies of the vaccines. This is now clearly mentioned in the revised ‘Materials and Methods’ chapter.

2) However - why no pilot study to enhance its robustness

Response: Thank you for the suggestion to discuss this. The study described here was only conducted for a short period at the beginning of the German COVID-19 vaccination campaign. The aim was to record the most frequent side effects of the vaccination regimens used at that time after basic immunisation in two different occupational groups that were to be recorded in a relatively well-defined manner. It became apparent, especially in the police sector, that access to this group was not easy. The results show that more research is certainly needed to address vaccination campaigns that are specifically targeted at different occupational groups, so that based on these results, further studies and cooperation can be initiated. Therefore, we believe that our work might well be considered as such a ‘pilot study’.

  1. C) Conclusion - you talk about communication strategies - who should instigate these/ follow these up given the appreciable extent of misinformation that exists with the Vaccines - and the need to overcome these? Good to give some guidance here

Response: Again, thank you very much for pointing out this problem. We have added a suggestion to the ‘Discussion’, which is in line with your comment: “The development of these communication and education strategies should first and foremost involve a panel of experts, including representatives from science, the medical services and possibly also from the communication science sector. This committee should then make a clear recommendation, which should be implemented accordingly by policymakers. In addition, misinformation should be addressed and corrected as quickly as possible in order to overcome vaccine hesitancy and potential concerns within the population with evidence-based, scientific arguments.”

Round 2

Reviewer 1 Report

The author has made revisions according to the review comments, and the conclusion is reasonable, providing evidences for people to understand and solve common problems encountered during the vaccination.